# Stimuli-responsive mechanically interlocked polymer wrinkles

Mengling Yang[1,2], Shuai Chen [1,2], Zhaoming Zhang[1], Lin Cheng [1], Jun Zhao [1], Ruixue Bai[1], Wenbin Wang[1], Wenzhe Gao[1], Wei Yu [1], Xuesong Jiang [1] ✉ & Xuzhou Yan [1] ✉

Artificial wrinkles, especially those with responsive erasure/regeneration behaviors have gained extensive interest due to their potential in smart applications. However, current wrinkle modulation methods primarily rely on network rearrangement, causing bottlenecks in in situ wrinkle regeneration. Herein, we report a dually cross-linked network wherein [2]rotaxane cross-link can dissipate stress within the wrinkles through its sliding motion without disrupting the network, and quadruple H-bonding cross-link comparatively highlight the advantages of [2]rotaxane modulation. Acid stimulation dissociates quadruple H-bonding and destructs network, swiftly eliminating the wrinkles. However, the regeneration process necessitates network rearrangement, making in situ recovery unfeasible. By contrast, alkaline stimulation disrupts host–guest recognition, and subsequent intramolecular motion of [2] rotaxane dissipate energy to eliminate wrinkles gradually. The always intact network allows for the in situ recovery of surface microstructures. The responsive behaviors of quadruple H-bonding and mechanical bond are orthogonal, and their combination leads to wrinkles with multiple but accurate responsiveness.

Wrinkles, a kind of patterned surfaces with various microstructures, are ubiquitous in nature. They manifest in phenomena like fingerprints[1,2] and cortexes[3,4], serving pivotal roles in life's essential functions such as augmenting friction[5,6] and enhancing cognitive capabilities[7]. Inspired by the exquisite and functional structures, artificial wrinkles have been extensively developed by designing bilayer polymer systems[8–12], thus enabling precise control over material surface properties. Thereinto, dynamic bonds have been commonly employed in polymeric wrinkles, which play crucial roles in creating intricate patterns and endowing them with adjustable behaviors through erasure/regeneration evolution. The reported dynamic chemistry is mainly based on disulfide bonds[13], reversible Diels-Alder reactions[14,15], photosensitive anthracene dimerization[16,17], boronic ester bonds[18], and multiple hydrogen bonds[19,20]. In terms of the intelligent regulation of wrinkles, they typically adhere to a common mechanism: The polymer networks undergo rearrangement through the exchange of dynamic covalent bonds or the dissociation/reassociation of non-covalent bonds, which results in the dissipation of the applied stress existed in the wrinkles and thus leads to the erasure of patterns. The network rearrangement mechanism has demonstrated its efficacy in erasing wrinkles; however, it also presents inherent limitations which it comes to wrinkle recovery. For example, upon removing the external stimuli, additional thermo-treatment is necessary to regenerate wrinkles due to the mismatch in moduli within the bilayer system[10,13]. Even so, the regenerated wrinkles are unable to maintain consistency with the original ones because of the randomness of the bond exchanges. Therefore, if a dynamic bond capable of dissipating energy stored in the wrinkles without necessitating the rearrangement of the network structure could be exploited, a different wrinkle regulation mechanism would be developed, thereby realizing the in situ adjustment of dynamic wrinkles. However, as of now, this objective has yet to be attained.

[1]School of Chemistry and Chemical Engineering, Frontiers Science Center for Transformative Molecules, Shanghai Jiao Tong University, Shanghai 200240, PR China. [2]These authors contributed equally: Mengling Yang, Shuai Chen. ✉e-mail: ponygle@sjtu.edu.cn; xzyan@sjtu.edu.cn

Mechanically interlocked molecules (MIMs) are made up of two or more molecular components entwined in space through mechanical bonds, which cannot be separated without breaking the covalent bonds involved[21–25]. The most well-known MIMs are rotaxanes[26,27] and catenanes[28–30]. Such sophisticated topological structure allows for abundant dynamic behaviors, which thus bestow unique features upon the resultant mechanically interlocked materials: i) The existence of supramolecular templates allows mechanical bonds to respond to various stimuli such as acid/base, force, and heat, which induces intramolecular movements to modulate the material's properties[31–37]. ii) The motion of mechanical bonds facilitates the dissipation of energy without requiring the separation of their components, which preserves the structural integrity and represents a unique mechanism for energy dissipation[38–40]. iii) The motion of mechanical bonds is confined within a specific range with perfect reversibility, thus ensuring the performance recovery of the resultant materials[41,42]. Based on above analyses, we envision that in situ erasure/regeneration regulation could be achieved for wrinkles featuring mechanical bonds with reversible intramolecular motions, which would represent a special wrinkle regulation mechanism that doesn't rely on the rearrangement/exchange of dynamic (non)covalent bonds. Nevertheless, the exploitation of employing mechanically interlocked structures for wrinkle regulation remains an uncharted territory.

Herein, we report the fabrication and in situ regulation of stimuli-responsive wrinkles based on the dually cross-linked mechanically interlocked network (MIN) in which the first cross-link of [2]rotaxane can dissipate stress within the wrinkles through its sliding motion without disrupting the network, and the second cross-link of quadruple H-bonding is employed to comparatively highlight the merits of mechanical bond in wrinkle regulation. In detail, the MIN with single mechanically interlocked cross-link which was composed of benzo-24-crown-8 (B24C8)-based [2]rotaxanes and nitrobenzyl-caged ureidopyrimidinone (NB-UPy) units[43], was firstly prepared on a polydimethylsiloxane (PDMS) substrate to produce a bilayer system. Subsequently, photo-induced deprotection of NB-UPy formed the quadruple H-bonding cross-link to enhance the modulus of the top MIN layer, resulting in the generation of wrinkles via the modulus mismatch mechanism (Fig. 1a). Under continuous stimulation with HCl, the MIN wrinkles were completely eliminated within 120 s due to rapid dissociation of the quadruple H-bonding, which could be regenerated by thermal treatment but with a changed surface microstructure (Fig. 1b, top). In contrast, Et₃N treatment also led to the elimination of wrinkles but continuous stimulation was not needed, and the erasure of wrinkles mainly occurred after removing the sample from the Et₃N atmosphere (Fig. 1b, bottom). This is because Et₃N only disrupted the host–guest interaction of [2]rotaxane, and the consequent motion of [2]rotaxane was responsible for energy dissipation. Intriguingly, the regenerated wrinkles upon thermal treatment closely matched the original ones, achieving an in situ recovery of the surface microstructure. Finally, the combination of [2]rotaxane-based mechanical bond and quadruple H-bonding integrates the antithetical characteristics of wrinkle regulations, encompassing acid/alkali responses, contact/non-contact responses, fast/slow responses, irreversible/reversible regeneration in a single system, showing fascinating dynamic properties.

## Results

### Design, synthesis, and structural characterization of the MINs

The mechanical bond adopted in our work is a [2]rotaxane, one of the most typical mechanically interlocked structures, known for its relatively facile synthesis and pronounced intramolecular motion. To better understand its role in wrinkle regulation, we combine [2]rotaxane with quadruple H-bonding to construct a dually cross-linked network rather than a network solely cross-linked by mechanical bond[44]. The reasons for this design are as follows: i) The inclusion of [2]

rotaxane and quadruple H-bonding in a single system allows for a direct comparison of their distinct erasure and regeneration mechanisms, which can highlight the uniqueness of mechanical motion in wrinkle regulation. ii) The two dynamic mechanisms are orthogonal and noninterfering, and have different responsive behaviors upon different stimuli, which endows corresponding wrinkles with multiple responsiveness. iii) The most fundamental factor is that quadruple H-bonding facilitates the formation of designable wrinkles. According to the mechanism, wrinkles occur to minimize the total energy of the bilayer films in cases where compressive strain exceeds critical strain. In other words, mismatch in moduli and thermal expansion ratios between the top layer and the substrate leads to the wrinkles[10,45,46]. Most simply, as long as bilayer films with different moduli are heated, the formation of wrinkles might be possible, but the controllability of the wrinkles is low. If the moduli of the top layer can be enhanced on demand, the wrinkles can be formed as designed. To achieve this, the UPy units, initially protected by a nitrobenzyl group, is copolymerized with a [2]rotaxane cross-linker to create a singly cross-linked network as the top layer (Fig. 1a). When exposed to UV irradiation at 365 nm, the UPy units are deprotected, forming a quadruple H-bonding dimer as the second cross-linking point that significantly enhances the modulus. This variation in modulus induces wrinkles formation, and the wrinkle patterns can be controlled by applying different photomasks on the top layer.

To establish the relationships between dynamic interactions and wrinkle regulation, the energy dissipation mechanisms of the two kinds of cross-linking points and their impacts on the mechanical properties of the MINs should be firstly confirmed. To this end, we prepared three MIN samples with different ratios of [2]rotaxane, NB-UPy, and butyl methacrylate (BMA) by free radical polymerization for the investigation of their structure–property relationships. The molar ratios of [2]rotaxane, NB-UPy, and BMA of MINs-1–3 were 0.75: 0.25: 100, 0.5: 0.5: 100, and 0.25: 0.75: 100, respectively. To better comprehend the dually cross-linking effect, we also prepared two control samples consisting of singly cross-linked networks. The molar ratios of [2]rotaxane, NB-UPy, and BMA in controls-1 and -2 were 1: 0: 100 and 0: 1: 100, respectively. Detailed information regarding these model networks can be found in Supplementary Table 1.

The structure of [2]rotaxane was initially confirmed by ¹H NMR. Compared to the pure B24C8 wheel, the aromatic proton H^i displayed a noticeable upfield shift, and the ethyleneoxy protons H^{2–7} on the wheel became complex (Fig. 2a and b). Simultaneously, in comparison with the spectrum of the vinyl-functionalized axle, the aromatic protons H^a and the methylene protons H^h on the axle shifted upfield, while the methylene protons H^f and H^i close to the secondary ammonium salt recognition site shifted downfield (Fig. 2b, c). All these results are consistent with the formation of host–guest complex. Furthermore, the methylene protons H^g, the nitrogen proton H^d and the aromatic protons H^b on the axle shifted downfield after the formation of [2]rotaxane, which supported the success of the stopping reaction (Fig. 2b, c). Meanwhile, the formation of NB-UPy was also confirmed by ¹H NMR and ¹³C NMR (Supplementary Figs. 22 and 23). To verify the success of the radical polymerization reaction, Fourier transform infrared (FT-IR) spectroscopy were conducted (Supplementary Fig. 43). The characteristic peak of C=C stretching vibration at 1640 cm⁻¹ was found in the FT-IR spectra of [2]rotaxane, NB-UPy and BMA, but it disappeared in the spectrum of MIN-2, indicative of the finished polymerization reaction.

### Thermal properties of the MINs

Thermal gravimetric analysis (TGA) was used to analyze the thermal stability of MINs. Both MINs and controls exhibited good thermal stability, with decomposition temperatures exceeding 255 °C at 95% weight (Supplementary Fig. 44). In addition, the glass transition temperatures ($T_g$) of the MINs with different cross-linking densities were

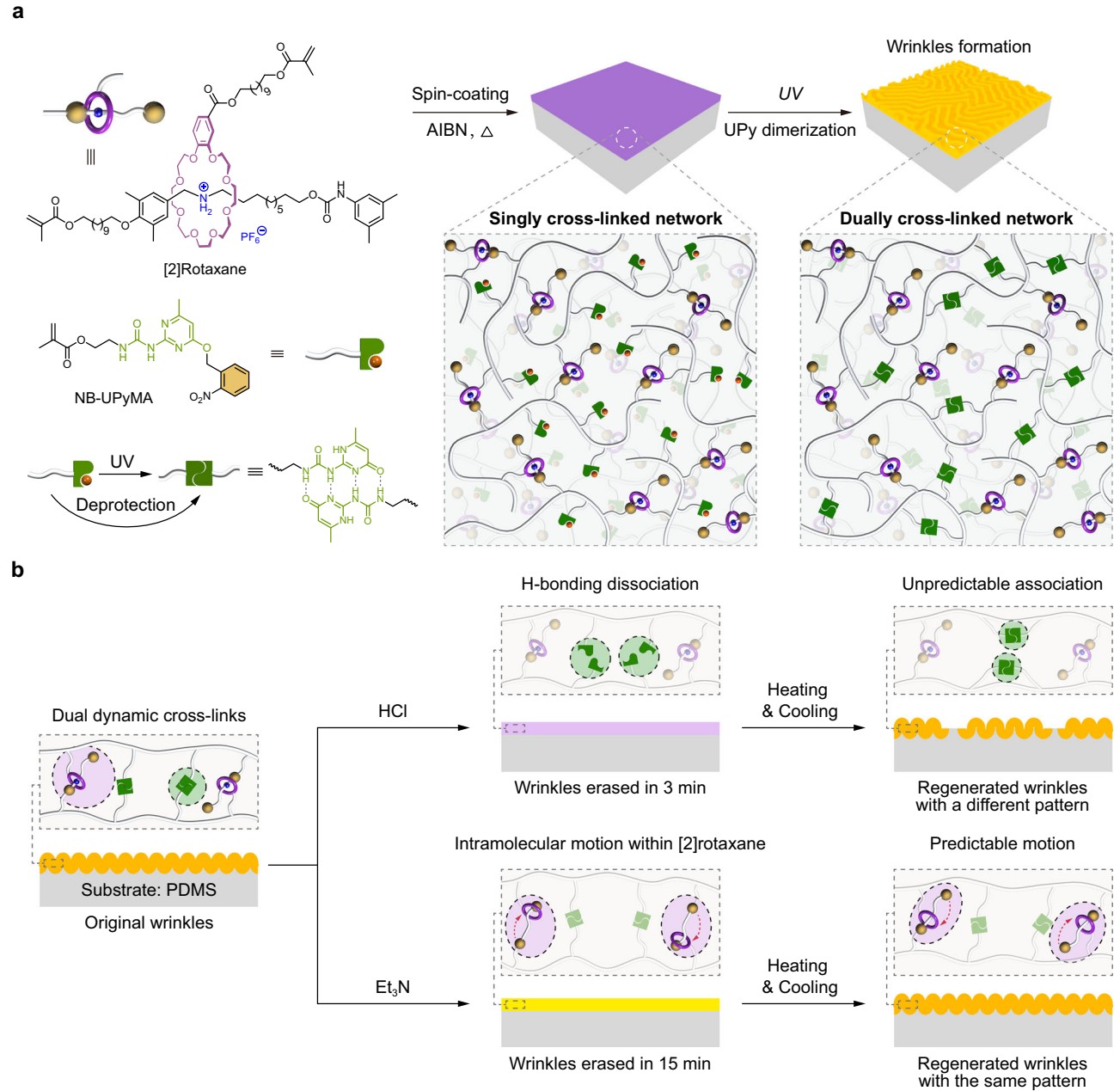

**Fig. 1 | The mechanism for stimuli-responsive wrinkles. a** Chemical structures of [2]rotaxane and NB-UPy as well as the schematic illustration of the formation of wrinkles based on the dually cross-linked MIN. **b** Schematic illustration of the two regulation pathways to eliminate and restore wrinkles based on a MIN/PDMS bilayer system.

determined by differential scanning calorimetry (DSC). The results showed that with the increase of [2]rotaxane content (corresponding to the decrease of UPy content), the $T_g$ value slightly decreased from 24.77 to 20.26 °C (Supplementary Fig. 45). It may be due to the [2] rotaxane cross-linker being slightly longer and more flexible compared to the quadruple H-bonding units, leading to a looser network and lower $T_g$.

**Mechanical properties of the MINs**

The fundamental mechanical properties of the model networks were evaluated using uniaxial tensile tests conducted at room temperature. The stress–strain curves, maximum stress values, and calculated Young's moduli for MINs-1, -2, and -3 are presented in Fig. 3a, b. It was observed that the maximum stress values were 5.1 MPa for MIN-1, 7.0 MPa for MIN-2, and 9.1 MPa for MIN-3. Additionally, the Young's moduli of MINs-1, -2, and -3 gradually increased from 9.1 MPa to

87.0 MPa and then to 175.8 MPa. Conversely, the maximum strains exhibited a gradual decrease from 440% to 371% and then to 298%. These results suggest that increasing the content of UPy units in MINs contributes to an enhancement in modulus and breaking stress[47]. However, the [2]rotaxane cross-linking structure provides better ductility performance, possibly originating from their different dynamic modes and molecular flexibility. Among the tested samples, MIN-2, with a 1:1 ratio of [2]rotaxane and UPy unit, demonstrated a more balanced set of properties. Consequently, it was selected for further investigation into the underlying mechanisms and the construction of responsive wrinkles.

The stress–strain curves of MIN-2 along with two control samples were obtained for comparison purposes (Fig. 3c). The corresponding data on maximum stress and Young's modulus were summarized in Fig. 3d. Control-1 exhibited the highest strain at break (478%), but the smallest maximum stress (3.8 MPa) and Young's modulus (1.1 MPa)

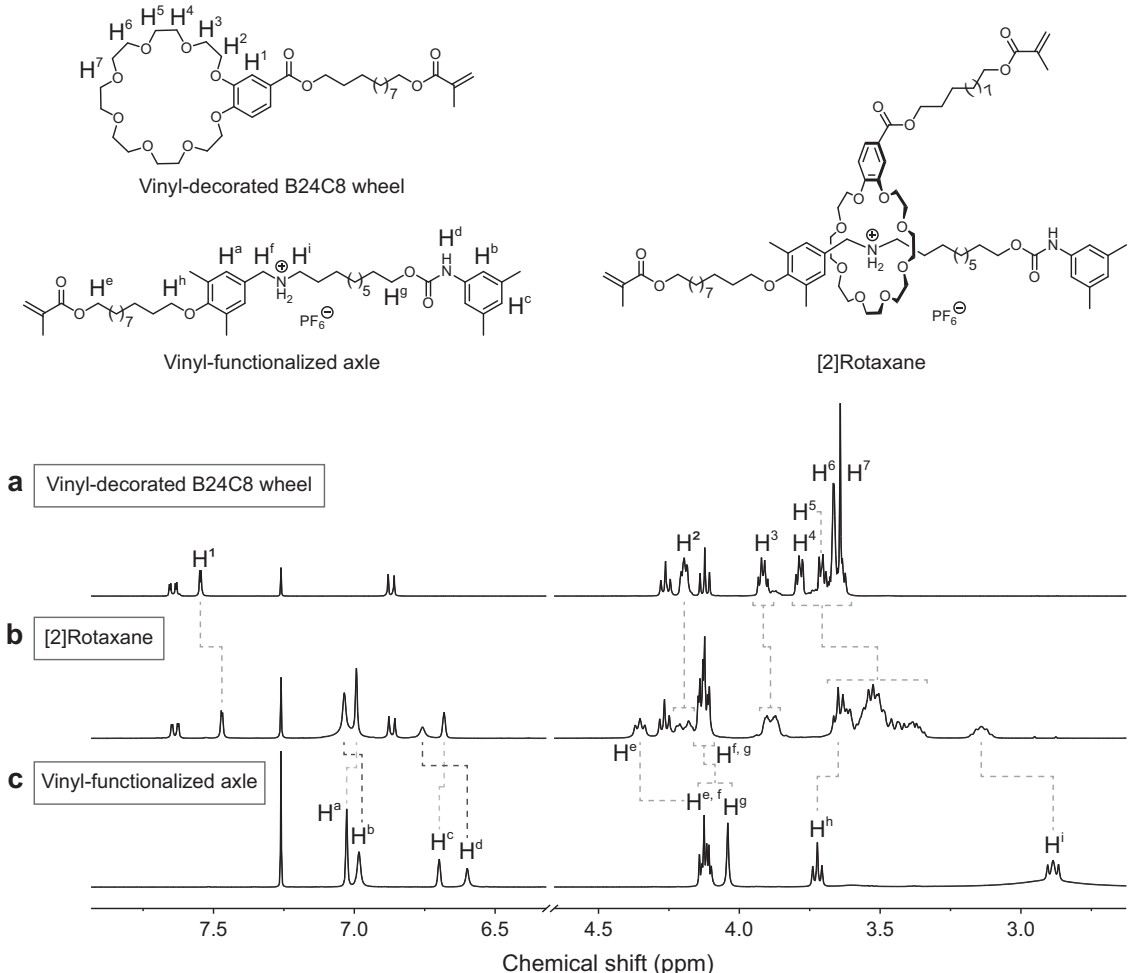

**Fig. 2 | Structural characterization of [2]rotaxane.** Partial $^1$H NMR spectra (400 MHz, CDCl$_3$, 25 °C) of (**a**), vinyl-decorated B24C8 wheel, (**b**) [2]rotaxane, and (**c**) vinyl-functionalized axle.

among all tested samples. Conversely, control-2 displayed completely opposite trends, with the lowest strain at break (158%), but the highest maximum stress (10.5 MPa) and Young's modulus (293.7 MPa). These trends observed in the controls align with the results obtained for MINs-1–3 and can be attributed to variations in their cross-links. Quadruple H-bonding has a rigid structure and is prone to forming hard phase aggregation, thereby enhancing the material's hardness and strength[47]. In contrast, the structure of [2]rotaxane is relatively flexible, and stretching can lead to its release of hidden lengths, which is advantageous for improving ductility and toughness. As for the MIN-2, the dually cross-linked strategy provides a synergistic effect, resulting in performance combination of the two kinds of cross-links.

Tensile tests were further conducted at various stretching rates (Fig. 3e). The results demonstrated that the mechanical behaviors of MIN-2 were highly dependent on the deformation rate. Both the yielding stress and Young's modulus of MIN-2 increased with increasing stretching rates ranging from 0.5 to 500 mm/min, underscoring notable dynamic characteristics (Supplementary Fig. 46). Furthermore, progressive cyclic tensile experiments were performed under different applied strains to evaluate the energy dissipation capability of MIN-2 (Fig. 3f and Supplementary Fig. 47). With increasing applied strains from 50% to 350%, the hysteresis loop expanded, and the residual strain became more pronounced. Additionally, the calculated values of energy dissipation exhibited an almost linear increase with different strains. The damping capacity, defined as the ratio of dissipated energy to incoming energy, consistently remained a high level

of averaging 90.3% across all applied strains. Similarly, controls-1 and -2, featuring single UPy and [2]rotaxane cross-links, respectively, also displayed distinct hysteresis in the cyclic tensile curves. The calculated damping capacities for controls-1 and -2 were found to be 80% and 95%, respectively (Supplementary Fig. 48). These findings suggest that both UPy and [2]rotaxane cross-links could effectively dissipate energy of the dynamic networks.

## Insights into the dynamic behaviors of the MINs

Rheology was employed to investigate the dynamic behaviors of hydrogen bonds and mechanical bonds at different time scales. Utilizing the time-temperature superposition (TTS) principle, master curves were obtained for MIN-2 and controls at a reference temperature of 50 °C (Fig. 4a–c). The master curves of MIN-2 can be divided into three regions based on the intersection of $G'$ and $G''$ (Fig. 4a). The high-frequency region above $f_g$ (>10$^3$ rad/s) with $G' > G''$ was the glassy regime where the network was frozen. The region below the crossover point ($f_d < f < f_g$) was identified as the transition region ($G' < G''$), where relaxation associated with the network strands occurred and was dominated by the $T_g$. Below the crossover point of $f_d$ ($G' > G''$), the master curve presents a plateau regime. The platform modulus of MIN-2 in the plateau regime was slightly higher than that of control-1 (Fig. 4b), but significantly lower than that of control-2 (Fig. 4c), aligning with the results of the tensile tests. Notably, control-2 exhibited a $G' / G''$ crossover point ($f_s$) at a low frequency around 10$^{-4}$ rad/s. At frequencies lower than $f_s$, a noticeable viscous flow

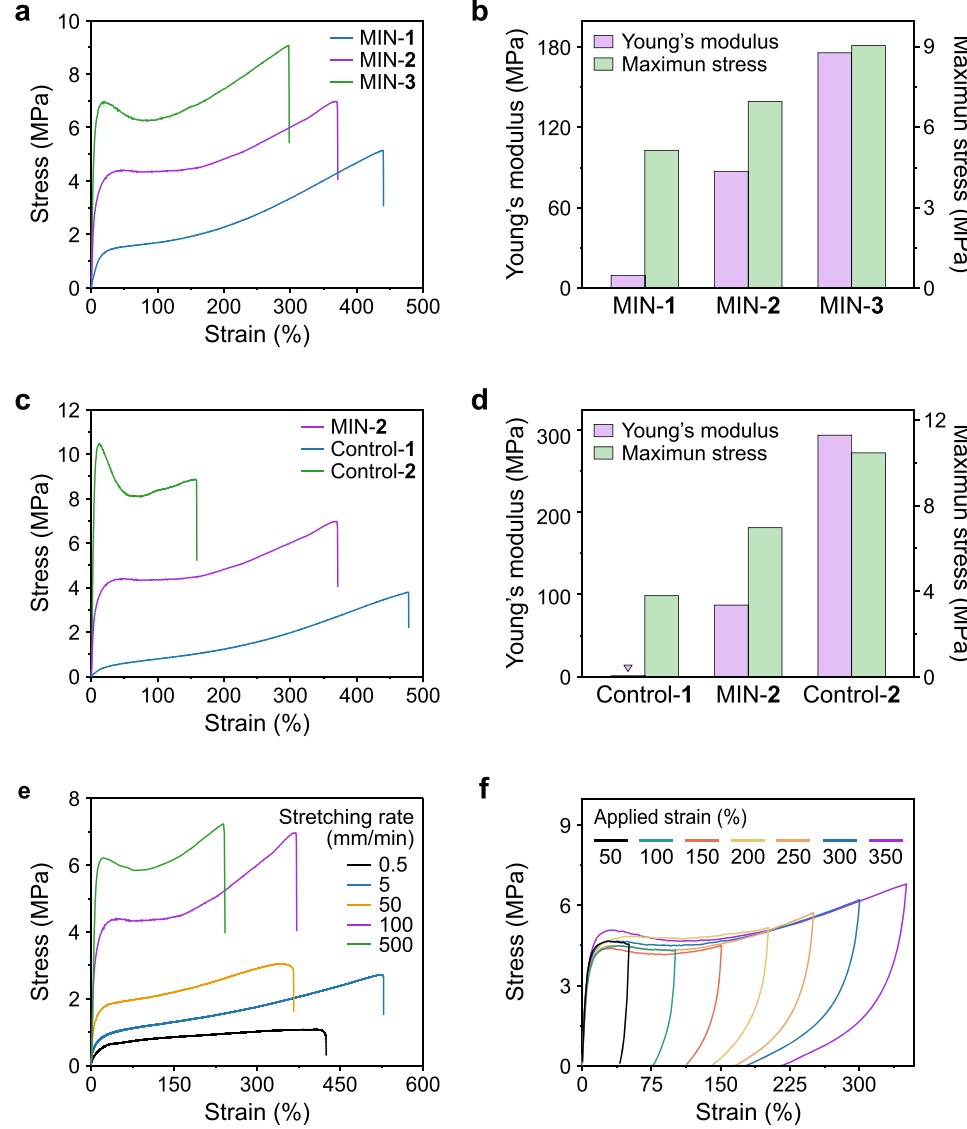

**Fig. 3 | Fundamental mechanical properties of MINs. a** Stress–strain curves of MINs recorded with a deformation rate of 100 mm/min. **b** Young's modulus and maximum stress of MINs calculated based on their stress–strain curves. **c** Stress–strain curves of MIN-2 and controls recorded with a deformation rate of 100 mm/min. **d** Young's modulus and maximum stress of MIN-2 and controls calculated based on their stress–strain curves. **e** Tensile stress–strain curves of MIN-2 at different stretching rates. **f** Cyclic tensile test curves of MIN-2 recorded with increased maximum strains.

characteristic was observed with $G' < G''$ (Fig. 4c). Typically, a viscous flow regime corresponds to the disruption of network structures. The viscous flow regime was not presented in the master curves of MIN-2 and control-1 (Fig. 4a, b). Additionally, the storage modulus of MIN-2 shows two plateaus, with a transition point around $10^{-4}$ rad/s, corresponding to hydrogen bond dissociation. These findings suggest that the dynamic network cross-linked by UPy units can be disrupted by external stimuli, while the network cross-linked by [2]rotaxane motifs remain intact. This observation aligns well with our speculation that the UPy cross-linked network dissipates energy through the destruction/reformation of the network, whereas network rearrangement cannot occur in the [2]rotaxane cross-linked network. Moreover, the peaks of tan $\delta$ curves at low frequency could be used to analyze the relaxation of the dynamic networks. In the tan $\delta$ curve of control-1, a small peak at around $10^{-5}$ rad/s could be differentiated (Fig. 4b), likely corresponding to the relaxation arising from the intramolecular motion of [2]rotaxane units.

Furthermore, we analyzed the activation energy for the dissociation of quadruple H-bonding ($E_{a,H}$) and the sliding motion of [2]rotaxane ($E_{a,slide}$) by two TTS methods[48,49] in the dynamic networks. The activation energy was calculated by fitting the temperature-dependent horizontal shift factors ($a_T$) curve using the Arrhenius Eq. (1):

$$\ln a_T = \ln A + \frac{E_a}{RT} \qquad (1)$$

Firstly, the master curves of MIN-2 were built under the guidance of high-$\omega$ $G''$ data, where the shift factors $a_{T,high}$ reflected the temperature dependence of the modulus contributed from the Rouse-type motion. The activation energy for the Rouse segmental motion ($E_{a,seg}$) was obtained from the temperature dependence of $a_{T,high}$ (purple symbols in Fig. 4d), resulting in $E_{a,seg} = 126.7$ kJ/mol. Subsequently, the master curves were established under the guidance of low-$\omega$ $G'$ data, where the shift factors $a_{T,low}$ encompassed the temperature dependence of the modulus from the dissociation of quadruple H-bonding and the intramolecular motion of [2]rotaxane. The activation energy calculated from the temperature dependence of $a_{T,low}$ was 164.6 kJ/

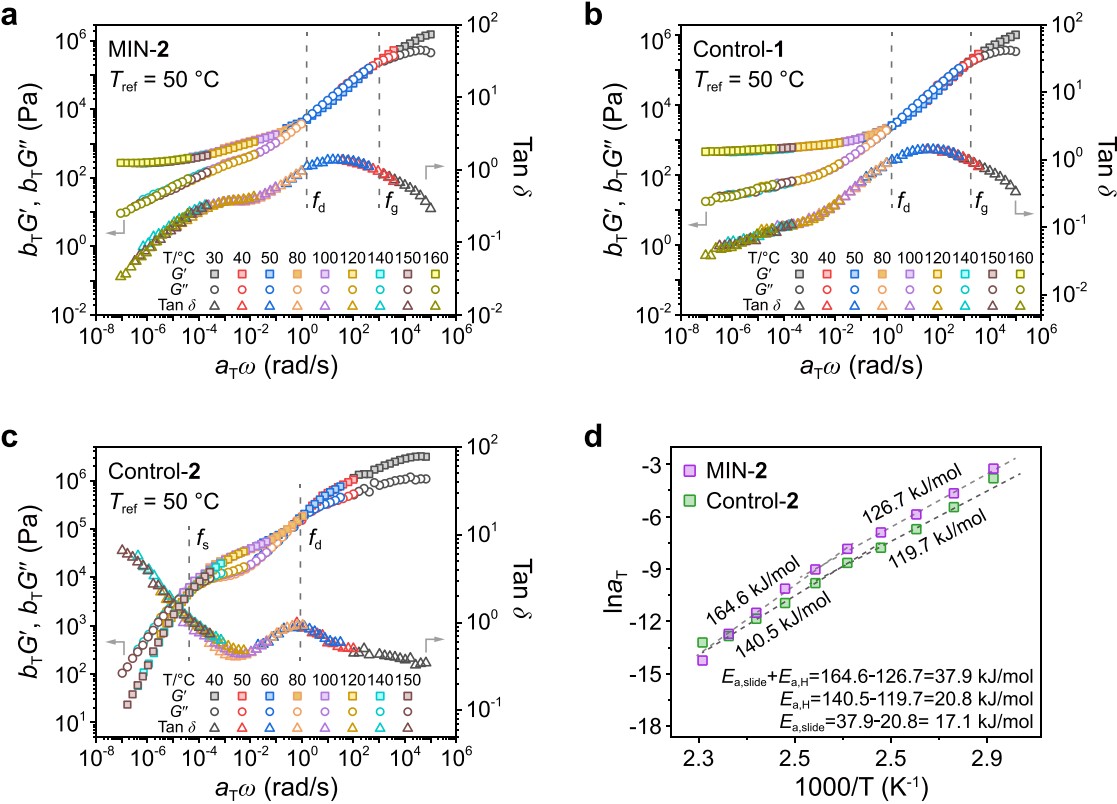

**Fig. 4 | Dynamic behaviors of MIN-2 and controls.** Master curves of (**a**), MIN-2, (**b**), control-1, and (**c**), control-2 at a reference temperature of 50 °C. **d** Dependence of the shift factors ($a_{T,high}$ and $a_{T,low}$) for MIN-2 and controls on the reciprocal of the temperature. The solid line and dotted line are the fitting lines of the Arrhenius equation.

mol, encompassing contributions from segmental motions ($E_{a,seg}$), dissociation of quadruple H-bonding ($E_{a,H}$), and sliding motion of [2] rotaxane ($E_{a,slide}$). By subtracting the $E_{a,seg}$ value of 126.7 kJ/mol, the combined activation energy of $E_{a,slide}$ and $E_{a,H}$ was determined to be 37.9 kJ/mol. To specifically elucidate the activation energy of $E_{a,H}$, we analyzed control-2 using the aforementioned method (green symbols in Fig. 4d). The $a_{T,high}$ and $a_{T,low}$ obtained by two TTS methods corresponded to $E_{a,seg}$ (119.7 kJ/mol) and the sum of $E_{a,seg}$ and $E_{a,H}$ (140.5 kJ/mol), respectively. Therefore, their difference could be attributed to the activation energy $E_{a,H}$, with a value of 20.8 kJ/mol. Utilizing this value for $E_{a,H}$, we further calculated the activation energy of $E_{a,slide}$ to be 17.1 kJ/mol. These results revealed the dynamic features of the two kinds of dynamic interactions, which provides a guidance to regulate the stimuli-responsive wrinkles.

### Preparation of wrinkles based on a MIN/PDMS bilayer system

In the preceding sections, the dynamics of [2]rotaxane and UPy cross-linking points in MIN has been well understood, then we aim to investigate their behaviors in wrinkle regulation. For this purpose, we first utilized the MIN with the two kinds of cross-linking points to fabricate the responsive wrinkled patterns (Fig. 5a). A top layer of MIN with nitrobenzyl-caged UPy was prepared on the bottom layer of PDMS substrate. Subsequently, upon UV light irradiation (365 nm) followed by heating and cooling treatment, random wrinkles generated via a modulus mismatch mechanism, which can be clearly exhibited by two-dimensional laser scanning confocal microscopy (LSCM) images (Fig. 5b) as well as corresponding three-dimensional images (Fig. 5c). Besides, the distinct light diffraction patterns also supported the formation of wrinkles with specific surface morphologies (Fig. 5b, insets). For the molecular mechanism, the removal of the photoprotective group in NB-UPy activates the formation of UPy dimers, causing the top layer of MIN becomes rigid (Fig. 5a). Due to the

mismatch in the modulus and thermal expansion ratio between PDMS substrate and the hardened MIN top layer, equi-biaxial compressive stress was generated upon cooling the heated bilayer to room temperature, which thus induced the formation of wrinkles as a result of releasing the localized stress to minimize the total energy of the system.

Particularly, achieving ordered wrinkles is possible by controlling the modulus distribution within the top layer, as shown in Fig. 5b(iii), c(iii). By covering the top layer with a photomask and exposing it to UV irradiation, wrinkles were selectively generated through the formation of quadruple H-bonding, while the unexposed area remained smooth. The wrinkles were oriented nearly perpendicular to the boundary of the exposed regions due to the boundary effect. Furthermore, finite element (FE) simulations were performed using Abaqus software (version 2020)[50] to analyze the formation process of wrinkles by setting the modulus change of the top layer before and after the photo-irradiation. In Supplementary Fig. 51, it can be observed that the stress distribution on the unexposed MIN/PDMS bilayer surface was uniform, without the formation of noticeable wrinkles. On the other hand, the stress distribution on the exposed surface (setting with increased modulus) was uneven, forming distinct wrinkles, which is in good agreement with the experimental results. All these results indicate that by controlling the stress distributions of the bilayer films based on dynamic chemistry, it is possible to customize surface patterns.

### Regulation of wrinkles based on a MIN/PDMS bilayer system

For the investigation of wrinkle regulation, their response to the dissociation/reassociation of UPy units was first examined. The wrinkled surface was exposed to an atmosphere with a partial pressure of HCl of approximately 90 Pa. In situ optical analysis revealed significant changes in the wrinkle pattern after 60 s of exposure, and within 180 s, the surface reached a wrinkle-free state (Fig. 6a(i–iii) and

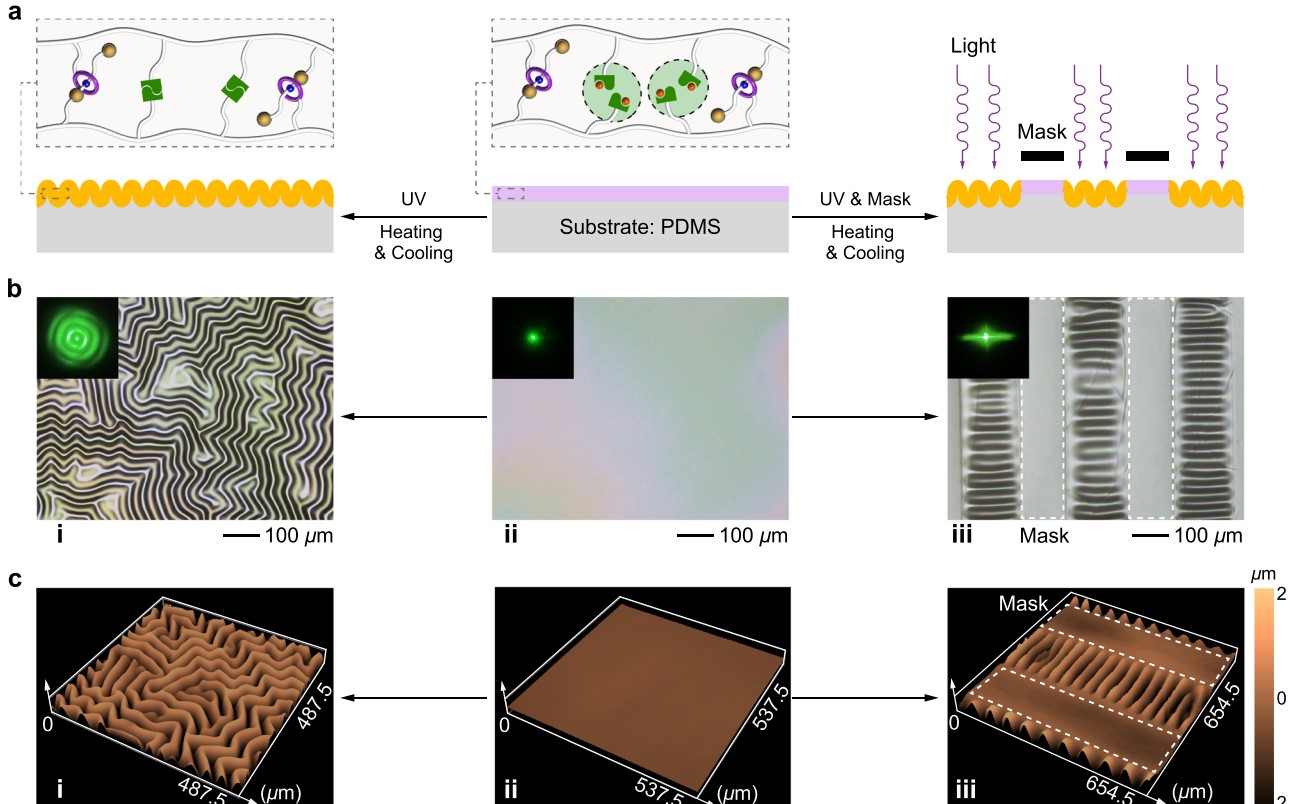

**Fig. 5 | Strategy for fabricating the responsive wrinkles. a** Schematic illustration of the formation of wrinkles based on a MIN/PDMS bilayer system. **b** 2D optical images and (**c**) 3D LSCM images of wrinkling surfaces on a MIN/PDMS bilayer upon sequential 365 nm UV irradiation and thermal treatment. The insets are the corresponding light grating patterns. The white dashed box indicates the unexposed region.

Supplementary Fig. 52a). These observations indicate that the wrinkle pattern exhibits highly responsive behavior towards HCl. To illustrate the changes in wrinkles more clearly, data on the characteristic amplitude ($A$) and wavelength ($\lambda$) were collected over time during the treatment with HCl vapor (Fig. 6c, e). Here, $A$ represents the height between peaks and valleys, while $\lambda$ refers to the distance between adjacent peaks in the wrinkles. Both of them decreased rapidly with the time treated with HCl vapor. The phenomena could be ascribed to the network rearrangement mechanism (Fig. 1b, top). When stimulated by HCl, the pyrimidine groups undergo protonation, leading to the dissociation of quadruple H-bonding interactions. As a result, the cross-linking density and modulus of the network decreased, which induced the release of internal compressive stress within the bilayer system. Consequently, the surface wrinkles are wiped away. After subsequent heating and cooling of the bilayer system, HCl was evaporated, and the quadruple H-bonding interactions reformed, resulting in the regeneration of wrinkles (Fig. 6a(iv)). However, the regenerated wrinkles are unpredictable and differ from the original ones depicted in Fig. 6a(i), because of the random rearrangement of the dynamic network.

Subsequently, the regulation of wrinkles by taking advantage of the intramolecular motion of [2]rotaxane was investigated (Fig. 6b). Initially, the motion of [2]rotaxane was hindered due to the stabilization of host−guest recognition, thus requiring disruption to initiate sliding motion. Herein, triethylamine (Et₃N) was used to neutralize the secondary ammonium salt motif and break the recognition. Unlike the HCl experiment, the sample was first exposed to an Et₃N atmosphere for 15 min, during which the wrinkles remained unchanged. However, upon removal from the Et₃N atmosphere, spontaneous erasure of wrinkles was observed in situ, and the elimination process was completed in approximately 15 min (Fig. 6b(i–iii)). This discovery highlights a non-contacting and relatively slow relaxation process in responsive

wrinkles. To verify that the relaxation process originates from the motion of [2]rotaxane, a control sample consisting only of quadruple H-bonding cross-links was prepared. Similar to the above procedure, the control sample was initially exposed to an Et₃N atmosphere for 15 min and then transferred to an Et₃N-free environment for observation for 30 min (Supplementary Fig. 53). No changes in the wrinkles were observed throughout the entire process, confirming the involvement of [2]rotaxane in the relaxation process.

The analysis of the data on amplitude and wavelength reflected the unique erasure behaviors arising from the [2]rotaxane compared to the sample treated with HCl (Fig. 6c–f). After treatment with Et₃N vapor, the $A$ of the wrinkles gradually decreased over time, while the $\lambda$ remained relatively unchanged (Fig. 6d, f). In contrast, both $A$ and $\lambda$ underwent significant changes upon HCl stimulation (Fig. 6c, e). These findings suggest that the erasure of wrinkles with [2]rotaxane is characterized by a gradual reduction in wrinkle height without disrupting the wrinkle texture. This distinctive erasure behavior also gives rise to a peculiar regeneration performance. After heating and cooling treatment, the regenerated wrinkles exhibited a remarkable resemblance to the original ones, accomplishing an in situ recovery (Fig. 6b(iv) and Supplementary Fig. 36b). These phenomena can be interpreted by a special energy dissipation mechanism: following the disruption of host−guest recognition, the ensuing motion of [2]rotaxane dissipates the force within wrinkles, leading to a reduction in wrinkle height. Critically, the network consistently maintains its integrity throughout this process, preserving the intricate texture of the wrinkle and facilitating its reversible recovery (Fig. 1b, bottom). The speculation could be verified by finite element (FE) simulations. To mimic the dynamic mechanisms of UPy and [2]rotaxane cross-links, corresponding simulations were based on setting changed and unchanged stress concentration points, respectively. The results illustrate that the

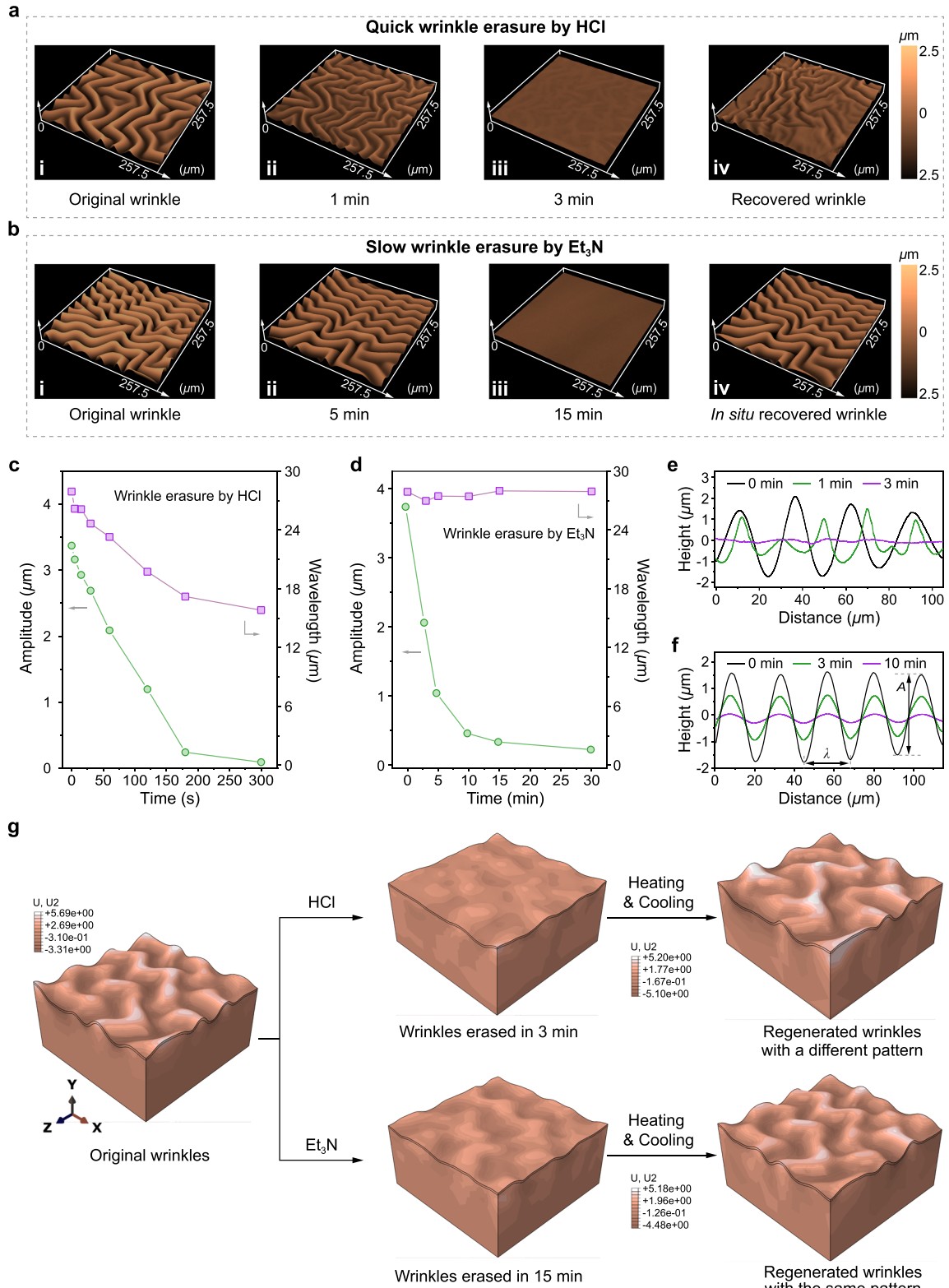

**Fig. 6 | Responsive wrinkles regulated by two different mechanisms.** 3D LSCM images of the erasure and recovery of wrinkling patterns via (**a**), sequential acid- and thermo-treatments and (**b**), sequential base- and thermo-treatments. *A* (green circle, left vertical axis) and *λ* (purple square, right vertical axis) of wrinkles change over time: (**c**), in HCl atmosphere and (**d**), after exposing the sample to Et₃N atmosphere for 15 min. The corresponding profiles of wrinkles after undergoing (**e**), HCl and (**f**), Et₃N treatment. **g** 3D FE simulation to compare erasure/regeneration evolution processes of responsive wrinkles upon HCl and Et₃N treatments.

morphology of wrinkles dominated by UPy dissociation exhibited significant changes after erasure and regeneration (Fig. 6g, top). In comparison, the morphology of wrinkles primarily controlled by the sliding motion of [2]rotaxane was capable of recovering to its original

patterns from a flat state (Fig. 6g, bottom), consistent with the experimental results.

From the above discussions, it was concluded that mechanical bonds can dissipate energy within wrinkles through intramolecular

motion to eliminate wrinkles, which is different from the network rearrangement mechanism in general dynamic wrinkles, representing a special wrinkle regulation mechanism. By simultaneously incorporating the dynamic interactions of quadruple H-bonding and mechanical bonds within a single system, the direct performance comparison reveals the features of the two kinds of dynamic interactions: i) In the case of commonly used dynamic interactions in wrinkles, such as quadruple H-bonding, specific stimuli can induce dynamic responsiveness, leading to alterations in network structures. Once the stimulus is removed, the response ceases immediately, requiring contact-based stimulation. Network disruption and rearrangement effectively dissipate stress, facilitating rapid wrinkle erasure. However, the resulting network structure differs from the original, making complete wrinkle recovery challenging. ii) In the case of mechanical bonds, the alkaline stimulus only disrupts the host–guest recognition, enabling intramolecular movement. Even after the stimulus is removed, the motion continues to release stress, allowing for non-contact stimulation. However, the efficiency of energy dissipation through intramolecular movement is lower than that of the network rearrangement, resulting in a slower wrinkle erasure. Throughout the erasure process, the network always remains intact, ensuring complete wrinkle recovery. Therefore, mechanical bonds not only overcome the limitations encountered by general dynamic bonds in wrinkle recovery but also exhibit erasing behaviors that are contrary to the dynamic networks, thus effectively complementing existing systems. Furthermore, the combination of these two classes of dynamic interactions enables the integration of original antithetical behaviors, including contact/non-contact responses, fast/slow responses, irreversible/reversible regeneration in a single system, which are orthogonal and noninterfering.

## Discussion

In summary, we designed and synthesized a dually cross-linked MIN based on a combination of [2]rotaxane and UPy quadruple H-bonding cross-linking points for the investigation of stimuli-responsive wrinkles. Fundamental tensile and rheological measurements revealed that both mechanical bonds and quadruple H-bonding are capable of dissipating energy. Thereinto, the [2]rotaxane cross-link achieves the dissipation through the sliding motion of mechanical bond, whereas the quadruple H-bonding is dependent on the dissociation of UPy dimer and destruction of the network structure. Coating the MIN with nitrobenzyl-caged UPy onto a PDMS substrate allows the preparation of a bilayer system. In this system, photoirradiation-induced deprotection of UPy leads to the formation of quadruple H-bonding cross-linking points, accompanied by an enhancement in the modulus of the top layer. This modulation of modulus promotes the formation of wrinkles, and the use of photomasks enables the creation of diverse surface wrinkle patterns. Importantly, the distinct energy dissipation mechanism of [2]rotaxane endows the wrinkles with different responsive behaviors compared to quadruple H-bonding. Similar to general dynamic bonds, acid stimulation triggers the dissociation of quadruple H-bonding, disrupting the network structure and swiftly eliminating surface wrinkles. However, the regenerative process involves network rearrangement, making the in situ recovery unfeasible. Conversely, after the host–guest recognition in [2]rotaxane being destroyed by alkali stimulation, its sliding motion gradually eliminated wrinkles while preserving the intact texture, thereby enabling an in situ recovery. Our research develops a special approach and mechanism to regulate wrinkles. As far as we know, this is the unique case of achieving wrinkle erasure and in situ regeneration based on dynamic chemistry of mechanical bond. Furthermore, the regulation involving mechanical bond is orthogonal to the general dynamic bonds, offering an opportunity to develop wrinkles with multiple stimuli responsiveness and diverse dynamic behaviors. We believe that the fundamental study presented here will deepen the understanding on the mechanism of wrinkle regulation and facilitate the application of dynamic wrinkles in smart and high-precision materials.

## Methods

### Preparation of the MINs and controls

In a typical procedure for preparation of MIN-2, [2]rotaxane (76.00 mg, 0.05 mmol), nitrobenzyl (NB)-caged 2-(2-ureido-4[1H]-6-methylpyrimidinone)ethyl methacrylate (NB-UPy, 21.00 mg, 0.05 mmol), butyl methacrylate (BMA, 1.42 g, 10.00 mmol) and DMF (1.50 mL) were added to a vial, bubbled with nitrogen gas at room temperature for 30 min, and stirred at room temperature until all monomers were completely dissolved. Then, azobis(isobutyronitrile) (AIBN, 17.00 mg, 0.10 mmol) was added to initiate the radical polymerization. The mixture was immediately transferred to a Teflon mold and kept at 70 °C for 48 h under $N_2$, and then dried under vacuum at 80 °C for 48 h to afford a film. The resultant film was further irradiated under 365 nm ultraviolet light for 7 h to yield MIN-1. MIN-2, and MIN-3. The controls were prepared through the same procedure and the monomer feedings for each sample were listed in Supplementary Table 1.

### Preparation and erasure of wrinkle pattern based on a MINs/PDMS bilayer

A mixture of tetrahydrofuran solution comprised of monomers (the molar ratio of [2]rotaxane and NB-UPy was 1:1) was spin-coated onto PDMS elastomer as the top layer with a thickness of approximately 1.2 $\mu$m. After kept at 70 °C for 48 h under $N_2$, the resultant film was further irradiated by 365 nm UV light with an intensity of 50 mW/cm$^2$ for 30 min, followed by heated at 100 °C for 5 min. Due to the considerable mismatch of thermal expansion ratio and modulus between the top stiff layer and the soft PDMS substrate, wrinkles occurred when the irradiated sample was cooled down to room temperature. The preparation of control group without [2]rotaxane was consistent with the above preparation method. As for erasure of the wrinkled patterns, the samples underwent HCl gas or Et$_3$N gas, followed by being observed by LSCM.

### Characterization methods

Nuclear magnetic resonance (NMR) spectra were recorded with a Bruker Avance DMX 400 spectrophotometer with use of the deuterated solvent as the lock and the residual solvent or TMS as the internal reference. High resolution mass spectra were obtained on a Bruker SolariX 7.0 T FT-ICR MS spectrometer. Fourier transform infrared (FT-IR) spectroscopy was performed on a Thermoscientific Nicolet 6700 FT-IR spectrometer at room temperature in the range of 550 - 4000 cm$^{-1}$. The thermal stability analysis was conducted using a TA Instruments Q500 thermogravimetric analyzer (TGA) under the nitrogen. Each sample (~7 mg) was heated from ambient temperature to 800 °C with a heating rate of 20.0 °C/min. Transition temperatures of materials determined on a TA Instruments Q2000 differential scanning calorimetry (DSC) under the nitrogen. Each sample (~10 mg) was analyzed by utilizing a heat/cool/heat cycle with a heating or cooling rate of 20 °C/min. Rheological experiments were carried out using a TA Instruments ARES G2 strain–controlled rheometer with an 8.0 mm parallel plate attachment. Surface patterns with responsive wrinkles were recorded by laser scanning confocal microscopy (LSCM, LEXT VK-X1000, Keyence, Japan).

### Mechanical tests

The mechanical properties of the MINs were measured using an Instron 3343 machine in standard stress–strain experiments. Young's modulus was determined from the initial slope of the stress–strain curves. Toughness was obtained by integrating the area under stress–strain curve. Energy dissipation was calculated by integrating

the area encompassed by the cyclic tensile curves. Damping capacity was defined as the ratio of the dissipated energy (the area encompassed by the loading and unloading curves) to the loading energy (the area encompassed by the loading curve).

## Data availability

The authors declare that the data supporting the findings of this study are available within the paper and its supplementary information files or the data are available from the corresponding authors on request. Source data are provided with this paper.

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

## Acknowledgements

X.Y. acknowledges the financial support of the NSFC/China (22122105 and 22071152), the NSF of Shanghai (22dz1207603), the Shuguang Program of Shanghai Education Development Foundation and Shanghai Municipal Education Commission (22SG11), and the Starry Night Science Fund of Zhejiang University Shanghai Institute for Advanced Study (SN-ZJU-SIAS-006). Z.Z. acknowledges the financial support of the NSFC/China (22101175). W.Y. acknowledges the financial support of the NSFC/China (52333001). X.J. acknowledges the financial support of the NSFC/China (52025032).

## Author contributions

X.Y. and X.J. supervised this research. X.Y. and M.Y. conceived the project. M.Y. carried out the synthesis and some characterizations of the materials. S.C. carried out some characterizations of the materials. L.C. carried out the rheological tests under the supervision of W.Y. Z.Z., J.Z., R.B., W.W., and W.G. helped with the manuscript preparation. The manuscript was written by M.Y., S.C., Z.Z. and X.Y. with contributions from all the coauthors.

## Competing interests

The authors declare no competing interests.
