## [Peer Review File · Nature Communications]

REVIEWER COMMENTS

Reviewer #1 (Remarks to the Author):

Artificial wrinkles, inspired by nature, have attracted significant attention in various fields such as cell biology, tissue engineering, optics, and electronics. Wrinkles with responsive erasure/regeneration behaviors enabled by dynamic chemistry further expand their intelligent applications. Yet, current wrinkle regulation methods mainly hinge on network rearrangement, facing limitations in in-situ regeneration post-erasure. In this manuscript, Yan and co-workers proposed an innovative wrinkle regulation mechanism, namely, introducing uniquely designed mechanical bonds to achieve in-situ erasure and regeneration of wrinkles, which represents a novel approach. Through the design of a dually cross-linked mechanically interlocked network, the authors successfully integrate [2]rotaxane and quadruple H-bonding cross-links, deeply exploring the differences between mechanical bond regulation and traditional methods. Under acidic and alkaline stimuli, these two cross-linking mechanisms exhibit distinct response behaviors. Impressively, the authors revealed the orthogonal nature of these two dynamic interactions, offering new possibilities for constructing multiple-response wrinkles. This research is significant for deepening the understanding of wrinkle regulation mechanisms and advancing the application of mechanically interlocked materials in surface patterning.

In summary, I consider this manuscript to be a good study. The authors provide detailed analysis and demonstration in structural design, material preparation, and performance characterization, offering readers rich information and profound insights. This manuscript is likely to be well-received by a broad readership. Therefore, the manuscript merits acceptance for publication in Nature Communications after addressing the following minor issues.

1. Previous studies have explored several physical methods for wrinkle regulation. How does the chemical regulation proposed in this manuscript differ from these approaches?
2. The dually cross-linked mechanically interlocked network proposed in the article has shown significant potential in dynamically regulating wrinkles. Can this mechanically interlocked structure be further optimized to enhance the efficiency and stability of wrinkle regulation?
3. The application of mechanically interlocked molecules in wrinkle regulation demonstrates greater potential for novel applications. However, what about the sustainability and difficulty of preparation of this novel material? How will future research address these challenges to achieve its widespread application?
4. When considering [2]rotaxane as the mechanically interlocked molecules for the research, did the authors contemplate other mechanically interlocked molecules? What prompted the decision to choose [2]rotaxane for investigating wrinkle regulation?
5. In this manuscript, it is mentioned that UV irradiation leads to the deprotection of UPy units, forming quadruple H-bonding dimers. What role does this step play in wrinkle formation? Additionally, are there alternative approaches to achieve comparable effects?

6. The "f1 (ppm)" of ¹H NMR spectrum in the Supplementary Fig. 21 in Supporting Information should be changed to "chemical shift (ppm)".

7. The statement "compound 7 is shown in Figure S21" in the line 221 of Supporting Information may be incorrect. It seems it is the compound 10 rather than compound 7. Please double-check it.

Reviewer #2 (Remarks to the Author):

Artificial wrinkles with specific topographies have gained extensive interest in fields of cell biology, tissue engineering, and the development of optics and electronics, and those with responsive erasure/regeneration behaviors further expand their smart applications. However, existing methods for modulating wrinkles are predominantly reliant on a network rearrangement mechanism, which presents performance bottlenecks, particularly in in situ wrinkle regeneration after erasure. In this nice manuscript, the authors report a dually cross-linked mechanically interlocked network wherein the [2]rotaxane cross-link can dissipate stress within the wrinkles through its sliding motion without disrupting the network structure, and the second cross-link of quadruple H-bonding comparatively highlight the advantages of mechanical bond modulation. On the one hand, acid stimulation triggers the dissociation of quadruple H-bonding and destruction of the network, swiftly eliminating the wrinkles. However, the regeneration process necessitates network rearrangement, making the in situ recovery unfeasible. On the other hand, alkaline stimulation disrupts host-guest recognition, and subsequent intramolecular motion of [2]rotaxane dissipate energy to eliminate wrinkles gradually. The always intact network allows for the in situ recovery of surface microstructures, representing a novel wrinkle regulation mechanism. The stimuli sources and responsive behaviors of regulation based on quadruple H-bonding and mechanical bond are orthogonal and noninterfering, and combining them in a single system leads to wrinkles with multiple but accurate responsiveness, showcasing smart behaviors. This nice manuscript, with good novelty and scientific value, can be published after the following minor revisions:

1. In the abstract, change "comparatively highlight the" to "comparatively highlights the".
2. In the abstract, change "motion of [2]rotaxane dissipate" to "motion of the [2]rotaxane units dissipate".
3. In the conclusion part, change "the fundamental study will deepen" to "the fundamental study presented here will deepen".

Reviewer #3 (Remarks to the Author):

The reconfigurable surface microstructure is intimately correlated with the microscale dynamic chemistry of polymer materials. In this work, the authors ingeniously utilized a dually cross-linked mechanically interlocked network to dynamically regulate surface wrinkles. Unlike previous methods, this approach enables the in-situ recovery of wrinkle morphology, presenting a novel

method for in-situ dynamic control of wrinkle morphology. Remarkably, different experimental phenomena arise from stimulation with triethylamine compared to stimulation with HCl, attributed to the acidic and basic stimulation acting on different crosslinking sites. This method verifies the multi-stimuli responsiveness of surface wrinkle patterns. Besides, I have several questions for the authors.

(1) The authors employed a dual cross-linking strategy to incorporate quadruple H-bonding into the mechanically interlocked network, directly comparing the characteristics of two dynamic interactions. However, apart from quadruple H-bonding, are there other dynamic bonds available for selection?

(2) The authors effectively regulated the surface wrinkles through acid-base stimulation. I am curious about the methods the authors used for sample treatment and real-time monitoring of the characteristic parameters of wrinkles (such as wavelength and amplitude). The authors should provide a detailed description of this process in the experimental section.

(3) The manuscript mentioned that alkaline stimulation disrupts host-guest recognition, enabling intramolecular motion of [2]rotaxane, which dissipates energy to achieve wrinkle erasure. Besides alkaline stimulation, is it possible to design other means of stimulation?

(4) In the erasure and recovery experiment of surface wrinkles induced by HCl, the results in Figure 6 indicate that it appears unable to fully recover to its original dimensions. What might be the reason for this? Could it be due to a change in the modulus of the system after HCl treatment, resulting in the inability to restore its morphology?

(5) In the manuscript, some of the fonts and font sizes should be uniform (e.g. Figure 3b-d, and Figure 6d). In Figure 5b, the exposed/unexposed region should be marked by some special symbols.

(6) In the support information (e.g., page s6, line 77, page s9, line 120), there are errors in the remarks of the nuclear magnetic instrument information, please verify and correct.

Overall, considering the demonstrated results with special phenomena and insightful mechanisms, this work is recommended for publication in Nature Communications.

Point-by-Point Response to Reviewers' Comments

For Reviewer 1:

1. Artificial wrinkles, inspired by nature, have attracted significant attention in various fields such as cell biology, tissue engineering, optics, and electronics. Wrinkles with responsive erasure/regeneration behaviors enabled by dynamic chemistry further expand their intelligent applications. Yet, current wrinkle regulation methods mainly hinge on network rearrangement, facing limitations in *in-situ* regeneration post-erasure. In this manuscript, Yan and co-workers proposed an innovative wrinkle regulation mechanism, namely, introducing uniquely designed mechanical bonds to achieve *in-situ* erasure and regeneration of wrinkles, which represents a novel approach. Through the design of a dually cross-linked mechanically interlocked network, the authors successfully integrate [2]rotaxane and quadruple H-bonding cross-links, deeply exploring the differences between mechanical bond regulation and traditional methods. Under acidic and alkaline stimuli, these two cross-linking mechanisms exhibit distinct response behaviors. Impressively, the authors revealed the orthogonal nature of these two dynamic interactions, offering new possibilities for constructing multiple-response wrinkles. This research is significant for deepening the understanding of wrinkle regulation mechanisms and advancing the application of mechanically interlocked materials in surface patterning.

In summary, I consider this manuscript to be a good study. The authors provide detailed analysis and demonstration in structural design, material preparation, and performance characterization, offering readers rich information and profound insights. This manuscript is likely to be well-received by a broad readership. Therefore, the manuscript merits acceptance for publication in Nature Communications after addressing the following minor issues.

- We would like to thank the reviewer for these valuable comments. The concerns raised by the reviewer have been elaborately addressed as shown below.

2. Previous studies have explored several physical methods for wrinkle regulation. How does the chemical regulation proposed in this manuscript differ from these approaches?

- We appreciate the reviewer's valuable and professional comment. As pointed by the reviewer, there are many classical works using physical methods to modulate the wrinkles. For example, common means include stretching (Rogers, J. A. et al. *Science* **2006**, *311*, 208–212; Gorodetsky, A. A. et al. *Adv. Mater.* **2020**, *32*, e1905717), swelling (Crosby, A. J. et al. *Adv. Mater.* **2006**, *18*, 3238–3242; **2011**, *23*, 4188–4192), and heating (Whitesides, G. M. et al. *Nature* **1998**, *393*, 146–149). The differences between our chemical regulation and them could be summarized as follows:

In terms of applicability, physical approaches typically offer a broader scope and do not have explicit structural requirements. Conversely, chemical methods often require polymer networks with liable chemical groups capable of responding to specific stimuli. In our system, these chemical groups include hydrogen bonds for acid response and the secondary ammonium salt for base response.

In terms of working mechanism, physical stimuli typically act on the entire system simultaneously. For instance, in the bilayer system, both layers are subjected to the same stimuli, and regulation success hinges on the distinct response behaviors of each layer. In contrast, our chemical regulation involves stimuli interacting solely with the top layer of the system. According to the typical linear buckling theory, the amplitude and wavelength of wrinkles can be adjusted by regulating the modulus of the top film in bilayer systems through dynamic chemistry, thus realizing the wrinkle regulations.

In terms of regulation behaviors, physical methods primarily affect the amplitude of wrinkles, with the wavelength generally remaining unchanged. In contrast, most chemical methods, including the HCl-triggered erasure in our work based on quadruple H-bonding, alter both the amplitude and wavelength of wrinkles due to network rearrangement triggered by chemical stimuli. Remarkably, the TEA-induced erasure behavior in our work is also a kind of chemical regulation based on the neutralization reaction. However, during this process, only the amplitude of the wrinkles changes, akin to physical methods, because the sliding motion of [2]rotaxane relaxes stress without disrupting the network structure. To our knowledge, the regulation based on [2]rotaxane represents the first chemical regulation which can only affect the amplitude of wrinkles and thus ensure an *in-situ* recovery of wrinkles.

3. The dually cross-linked mechanically interlocked network proposed in the article has shown significant potential in dynamically regulating wrinkles. Can this mechanically interlocked structure be further optimized to enhance the efficiency and stability of wrinkle regulation?

➤ We thank the reviewer for the comment. The wrinkle erasure mechanism in mechanically interlocked networks (MIN) relies on the movement of mechanical bonds to relax stress in the system. The motion feature of mechanical bonds could be manipulated by various structural factors. Consequently, we believe that the efficiency and stability of wrinkle regulation for MIN could be adjusted by the following approaches: (1) Adjusting the strength of supramolecular interactions within mechanical bonds. For instance, as depicted in Fig. I, replacing the host-guest recognition with stronger metal-coordination interactions can increase the activation threshold for motion, thereby enhancing the stability of the wrinkles. Alternatively, reducing the strength of these interactions can accelerate the dissociation rate of crown ethers, resulting in more sensitive erasure behavior. (2) Extending the length of the axle. Increasing the motion range of mechanical bond allows for greater energy dissipation through the release of a longer hidden length. Hence, the efficiency can be improved by designing the length of the axle to adjust the sliding distance of [2]rotaxane, as depicted in Fig. II. (3) Changing the motion mode of mechanical bonds. Different mechanically interlocked structures exhibit varied motion modes, which have different performance in energy dissipation. For example (Fig. III), [c2]daisy chain consist of two monomers where the guest of one monomer is complexed by the host of the second monomer through host-guest interactions and vice versa. The topological structure allows each component to slide past its partner in one dimension, which is reminiscent of filament sliding in sarcomeres. Additionally, catenanes, consisting solely of mechanically interlocked macrocycles, offer a wider range of conformational movements—including

circumrotation, rocking, and elongation—compared to rotaxanes. Therefore, [c2]daisy chain and [2]catenane can be used to optimize the dynamic behaviors of MIN wrinkles.

Fig. I. The proposed strategy of adjusting the strength of supramolecular interactions within mechanical bonds (Top: [2]rotaxane base on metal-coordination interaction; Bottom: [2]rotaxane base on BPP34C10 host and complementary paraquat guest).

Fig. II. [2]Rotaxanes with the axles of different lengths.

Fig. III. a, Schematic diagram of two [c2]daisy chains with different conformations and their force-responsive conformation changes (Yan, X. et al. *J. Am. Chem. Soc.* **2023**, *145*, 567–578). b, Cartoon representation of [2]catenane with the force-induced multiple motion patterns (Yan, X. et al. *J. Am. Chem. Soc.* **2023**, *145*, 9011–9020).

4. The application of mechanically interlocked molecules in wrinkle regulation demonstrates greater potential for novel applications. However, what about the sustainability and difficulty of preparation of this novel material? How will future research address these challenges to achieve its widespread application?

➤ We thank the reviewer for the valuable comments. Regarding the preparation of mechanically interlocked polymer materials, the key lies in the synthesis of mechanically interlocked molecules, with the critical step in the synthetic route being the final recognition end-capping reaction to form the target [2]rotaxane monomer. Thereinto, the syntheses of the starting materials of macrocyclic host **1** and secondary ammonium salt guest **2** were straightforward, which can be obtained in a relatively large-scale according to the established methods (Robin D. R. et al. *J. Am. Chem. Soc.* **1999**, *121*, 11281–11290; Coutrot et al. *J. Org. Chem.* **2010**, *75*, 6516–6531). Benefitting from the high affinity of the B24C8 host for the ammonium salt guest, the yield of the [2]rotaxane monomer can also reach up to 61%. Therefore, although the mechanically interlocked structures look very complex, they still can be prepared with sufficient quantity for applications in certain fields.

To achieve their sustainable utilization, one approach is to integrate dynamic chemistry into the synthesis of mechanically interlocked wrinkling materials. In our previous work, we have already worked on the synthesis of sustainable mechanically interlocked materials. For instance, we prepared the acetoacetate-decorated [2]rotaxane that undergoes catalyst-free condensation reaction with two commercially available multi-amine monomers to furnish mechanically interlocked vitrimers. By virtue of the vitrimer chemistry of vinylogous urethanes, we impart reprocessability and chemical recyclability to the mechanically interlocked materials (*J. Am. Chem. Soc.* **2022**, *144*, 872–882). Therefore, the combination of mechanical bonds and dynamic covalent bonds are applicable to solve the sustainability of mechanically interlocked materials.

In addition, we also intend to introduce the anthracene groups onto mechanically interlocked molecules, and then leverage photochemistry to address the challenge in sustainability of mechanically interlocked materials in the future research. The reversible dimerization of anthracene also makes it possible to the sustainable application of the mechanically interlocked materials. Currently, we are prepared to synthesize the anthracene-functionalized [2]rotaxane as depicted in Fig. IV, which will be utilized for application exploration in forthcoming studies.

Fig. IV. The proposed synthesis route of anthracene-modified [2]rotaxane and the dimerization and dedimerization mechanism of the photo-responsive mechanically interlocked material.

5. When considering [2]rotaxane as the mechanically interlocked molecules for the research, did the authors contemplate other mechanically interlocked molecules? What prompted the decision to choose [2]rotaxane for investigating wrinkle regulation?

➤ We thank the reviewer for the valuable comments. As interpreted above, as long as the mechanically interlocked structures could undergo effective motion upon stimuli to dissipate energy, they can be adopted to construct dynamic wrinkles. In this case, there are also many other good candidates such as [c2]daisy chain and [2]catenane. However, compared with [2]rotaxane, their structures are more complex and their synthetic processes are more intricate. [2]Rotaxane offers a more simple structure which guarantee an easier synthesis. Furthermore, the simple structure also indicates that it is also suitable to serve as a model structure to reveal the working mechanism of mechanically interlocked materials in wrinkle regulation (Yan, X. et al. *Nat. Commun.* **2022**, *13*, 6654; Takata, T. et al. *Angew. Chem. Int. Ed.* **2019**, *58*, 2765–2768). Therefore, we chose [2]rotaxane as the mechanically interlocked structure in our research.

6. In this manuscript, it is mentioned that UV irradiation leads to the deprotection of UPy units, forming quadruple H-bonding dimers. What role does this step play in wrinkle formation? Additionally, are there alternative approaches to achieve comparable effects?

➤ We thank the reviewer for the comments. UV irradiation induces the deprotection of UPy units, leading to the formation of quadruple hydrogen-bonding dimers. This process plays a crucial role in increasing the modulus of the top layer in the bilayer wrinkle system. According to classical bilayer wrinkle theory, the mismatch in modulus and thermal expansion ratio between the PDMS substrate and the hardened MIN top layer leads to equi-biaxial compressive stress upon cooling the heated bilayer to room temperature, which thus induced the formation of wrinkles as a result of releasing the localized stress to minimize the total energy of the system. Therefore, the formation of the wrinkles benefits from the deprotection of UPy units and subsequent formation of quadruple H-bonding dimers.

In addition to UPy, many other reactions can also be used to form wrinkle in bilayer systems. Similarly utilizing light stimulation, anthracene groups exhibit good reversible photo-dimerization properties. Researchers have introduced anthracene dimerization into the top layer of the bilayer system to construct dynamic wrinkles. Under UV light at 365 nm, the anthracene groups undergo photo-dimerization, causing the supramolecular network to cross-link and increase the modulus, which spontaneously generates wrinkle patterns on the surface. Due to the reversibility of the anthracene dimerization reaction, under UV light at 254 nm, the supramolecular network decrosslinks, the modulus decreases, and the wrinkle patterns disappear, creating a UV-responsive dynamic wrinkle pattern. (Yi, G. et al. *Nano Res.* **2023**, *16*, 634–644). Besides, thermal-induced reactions can also be adopted. Our previous work used PDMS as the substrate and a mixture of furan and maleimide as the top layer material (*Adv. Mater.* **2016**, *28*, 9126–9132). Upon heating the sample at 70 °C, a Diels-Alder cross-linking reaction occurred in the top layer, leading to an increase in modulus. This resulted in significant differences in modulus and thermal expansion coefficient between the top layer and the bottom PDMS substrate. Upon cooling to room temperature, contraction stress within the system spontaneously generated wrinkled patterns on the surface to minimize energy. Therefore, there are many alternative approaches to achieve similar effect as long as the reactions are effective enough upon specific stimuli.

7. The "f1 (ppm)" of ¹H NMR spectrum in the Supplementary Fig. 21 in Supporting Information should be changed to "chemical shift (ppm)".
- We thank the reviewer for the kind reminder of this point. We have addressed the issue by correcting "f1 (ppm)" to "chemical shift (ppm)" in Supplementary Fig. 21 on page S19. The corrected picture was also shown below.

Fig. V. (replacing Supplementary Fig. 21) ¹H NMR spectrum (CDCl₃, room temperature, 400 MHz) of compound **10**.

8. The statement "compound 7 is shown in Figure S21" in the line 221 of Supporting Information may be incorrect. It seems it is the compound 10 rather than compound 7. Please double-check it.
 - We thank the reviewer for the kind reminder of this point. We have thoroughly checked the Supporting Information and have rectified "compound 7 is shown in Figure S21" to "compound 10 is shown in Figure S21" on page S18.

For Reviewer 2:

1. Artificial wrinkles with specific topographies have gained extensive interest in fields of cell biology, tissue engineering, and the development of optics and electronics, and those with responsive erasure/regeneration behaviors further expand their smart applications. However, existing methods for modulating wrinkles are predominantly reliant on a network rearrangement mechanism, which presents performance bottlenecks, particularly in *in-situ* wrinkle regeneration after erasure. In this nice manuscript, the authors report a dually cross-linked mechanically interlocked network wherein the [2]rotaxane cross-link can dissipate stress within the wrinkles through its sliding motion without disrupting the network structure, and the second cross-link of quadruple H-bonding comparatively highlight the advantages of mechanical bond modulation. On the one hand, acid stimulation triggers the dissociation of quadruple H-bonding and destruction of the network, swiftly eliminating the wrinkles. However, the regeneration process necessitates network rearrangement, making the *in-situ* recovery unfeasible. On the other hand, alkaline stimulation disrupts host-guest recognition, and subsequent intramolecular motion of [2]rotaxane dissipate energy to eliminate wrinkles gradually. The always intact network allows for the *in-situ* recovery of surface microstructures, representing a novel wrinkle regulation mechanism. The stimuli sources and responsive behaviors of regulation based on quadruple H-bonding and mechanical bond are orthogonal and noninterfering, and combining them in a single system leads to wrinkles with multiple but accurate responsiveness, showcasing smart behaviors. This nice manuscript, with good novelty and scientific value, can be published after the following minor revisions:
 - We greatly appreciate the reviewer's profound evaluation of our work. The concerns raised by the reviewer have been elaborately addressed as shown below.
2. In the abstract, change "comparatively highlight the" to "comparatively highlights the".
 - We thank the reviewer for pointing this out. Accordingly, we have rectified the error by changing "comparatively highlight the" to "comparatively highlights the" in the abstract of the main text on page 1.
3. In the abstract, change "motion of [2]rotaxane dissipate" to "motion of the [2]rotaxane units dissipate".
 - We thank the reviewer for pointing this out. As suggested, we have corrected the "motion of [2]rotaxane dissipate" into "motion of the [2]rotaxane units dissipate" in the abstract of the main text on page 1.

4. In the conclusion part, change “the fundamental study will deepen” to “the fundamental study presented here will deepen”.

We thank the reviewer for pointing this out. As suggested, we have changed “the fundamental study will deepen” to “the fundamental study presented here will deepen” in the conclusion part of the main text on page 18.

For Reviewer 3:

1. The reconfigurable surface microstructure is intimately correlated with the microscale dynamic chemistry of polymer materials. In this work, the authors ingeniously utilized a dually cross-linked mechanically interlocked network to dynamically regulate surface wrinkles. Unlike previous methods, this approach enables the *in-situ* recovery of wrinkle morphology, presenting a novel method for *in-situ* dynamic control of wrinkle morphology. Remarkably, different experimental phenomena arise from stimulation with triethylamine compared to stimulation with HCl, attributed to the acidic and basic stimulation acting on different crosslinking sites. This method verifies the multi-stimuli responsiveness of surface wrinkle patterns. Besides, I have several questions for the authors.

➤ We are grateful for the reviewer’s comprehensive summary of the highlights in our work. The concerns raised by the reviewer have been elaborately addressed as shown below.

2. The authors employed a dual cross-linking strategy to incorporate quadruple H-bonding into the mechanically interlocked network, directly comparing the characteristics of two dynamic interactions. However, apart from quadruple H-bonding, are there other dynamic bonds available for selection?

➤ We thank the reviewer for the valuable comment. Besides quadruple H-bonding, there are many dynamic bonds available for selection, including but not limited to Diels-Alder reversible reaction (Jiang, X. et al. *Adv. Mater.* **2020**, *32*, 1906712), photo-reversible dimerization reaction (Yi, G. et al. *Nano Res.* **2023**, *16*, 634–644), reversible formation and cleavage of boronic ester bond (Jiang, X. et al. *Acta Polym. Sin.* **2021**, *52*, 61–68), reversible formation and cleavage of disulfide bond (Jiang, X. et al. *Nat. Commun.* **2022**, *13*, 7434), reversible formation and cleavage of hydrogen bond (Chen, X. et al. *Adv. Mater.* **2024**, 2314201), and so on. Researchers have demonstrated that by incorporating these dynamic bonds into the bilayer system, they can create modulus-tunable dynamic cross-linked polymer networks as the top layer, thus achieving various dynamic wrinkle patterns. Therefore, they can also theoretically be used as effective dynamic cross-links to construct dual cross-linked networks.

Notably, compared to the aforementioned dynamic bonds, quadruple H-bonding offers unique advantages. The dissociation or exchange reactions of these dynamic bonds typically rely on thermal treatment, which might inadvertently influence the motion of [2]rotaxane due to high temperatures. However, the stimulus responsiveness of quadruple H-bonding does not interfere with that of mechanical bonds. Moreover, its capacity for photo-induced formation facilitates wrinkle generation, which is also indispensable for the

cross-linked network. Therefore, quadruple H-bonding is more suitable for the comparison with [2]rotaxane, leading to its selection for constructing our dual cross-linked networks.

3. The authors effectively regulated the surface wrinkles through acid-base stimulation. I am curious about the methods the authors used for sample treatment and real-time monitoring of the characteristic parameters of wrinkles (such as wavelength and amplitude). The authors should provide a detailed description of this process in the experimental section.

➤ We thank the reviewer for the professional reminder of this point. As suggested, we have added the following detailed description in preparation and erasure of wrinkle pattern based on a MINs/PDMS bilayer part of the supplementary information on page S24.

Monitoring the amplitude (A) and wavelength (λ) of responsive wrinkles in real-time:

HCl: The wrinkled pattern was placed in a device filled with hydrogen chloride (HCl) atmosphere. After treatment for different times, the sample was observed by the three-dimensional laser scanning confocal microscopy (LSCM) for real-time data collection.

Et₃N: The wrinkled pattern was placed in a device filled with triethylamine atmosphere. After treatment for different times, the sample was observed by LSCM for real-time data collection.

4. The manuscript mentioned that alkaline stimulation disrupts host-guest recognition, enabling intramolecular motion of [2]rotaxane, which dissipates energy to achieve wrinkle erasure. Besides alkaline stimulation, is it possible to design other means of stimulation?

➤ We appreciate the reviewer for the valuable comment. In our work, we utilized host-guest recognition as a supramolecular template to construct mechanically interlocked molecule. By disrupting host-guest recognition through alkaline stimulation, it is possible to drive intramolecular motion to achieve wrinkle elimination. Here, the host-guest recognition acts as a gating unit, determining the intramolecular motion of mechanically interlocked molecule. Therefore, by altering the type of gating units, different modes of stimulation can be achieved. For example, if we use azobenzene as the gating unit for mechanically interlocked molecule, we can achieve light-responsive regulation. The scheme of such molecules is shown in Fig. VI. Additionally, metal-ligand coordination bonds and hydrogen bonds are also commonly used as synthetic templates to construct mechanically interlocked molecules. By utilizing these interactions to design gating units, different types of stimulations can be employed to regulate wrinkles.

Fig. VI. The proposed chemical structures and photoisomerization scheme of [2]rotaxanes with azobenzene derivative as the axle.

5. In the erasure and recovery experiment of surface wrinkles induced by HCl, the results in Figure 6 indicate that it appears unable to fully recover to its original dimensions. What might be the reason for this? Could it be due to a change in the modulus of the system after HCl treatment, resulting in the inability to restore its morphology?
 - We thank the reviewer for the comments. In the erasure and recovery experiments of surface wrinkles induced by HCl (Fig. 6), the results indicate that the surface wrinkles cannot fully recover to their original size, mainly due to the network rearrangement mechanism triggered by quadruple H-bonding. When stimulated by HCl, the pyrimidine groups of UPy unit could be protonated, and thus lead to the dissociation of quadruple H-bonding interactions. Accordingly, the cross-linking density and modulus of the network decreased, further resulting in the release of internal pressure within the bilayer system and the erasure of the surface wrinkle. After performing heating and cooling treatments on the bilayer system, HCl was evaporated, and the quadruple H-bonding interactions reformed to induce the regeneration of wrinkles. However, due to the random rearrangement of the dynamic network, the resulting network structure differs from the original, making complete recovery of wrinkling morphologies challengeable. It is for this reason that we have designed the mechanical bond regulation system, which can fully achieve the recovery of morphology.

6. In the manuscript, some of the fonts and font sizes should be uniform (e.g. Figure 3b-d, and Figure 6d). In Figure 5b, the exposed/unexposed region should be marked by some special symbols.
 - We thank the reviewer for pointing these out. We have standardized the fonts and font sizes in Figs. 3 and 6. Additionally, we have marked the exposed/unexposed regions to enhance the clarity of the picture in Fig. 5. The corresponding pictures are also shown below.

Fig. VII. (replacing Fig. 3) Fundamental mechanical properties of MINs. a, Stress–strain curves of MINs recorded with a deformation rate of 100 mm/min. b, Young’s modulus and maximum stress of MINs calculated based on their stress–strain curves. c, Stress–strain curves of MIN-2 and controls recorded with a deformation rate of 100 mm/min. d, Young’s modulus and maximum stress of MIN-2 and controls calculated based on their stress–strain curves. e, Tensile stress–strain curves of MIN-2 at different stretching rates. f, Cyclic tensile test curves of MIN-2 recorded with increased maximum strains.

Fig. VIII. (replacing Fig. 5) Strategy for fabricating the responsive wrinkles. a, Schematic illustration of the formation of wrinkles based on a MIN/PDMS bilayer system. b, 2D optical images and c, 3D LSCM images of wrinkling surfaces on a MIN/PDMS bilayer upon sequential 365 nm UV irradiation and thermal treatment. The insets are the corresponding light grating patterns. The white dashed box indicates the unexposed region.

Fig. IX. (replacing Fig. 6) Responsive wrinkles regulated by two different mechanisms. 3D LSCM images of the erasure and recovery of wrinkling patterns via a, sequential acid- and thermo-treatments and b, sequential base- and thermo-treatments. A (green circle, left vertical axis) and λ (purple square, right vertical axis) of wrinkles change over time: c, in HCl atmosphere and d, after exposing the sample to Et₃N atmosphere for 15 min. The corresponding profiles of wrinkles after undergoing e, HCl and f, Et₃N treatment. g, 3D FE simulation to compare erasure/regeneration evolution processes of responsive wrinkles upon HCl and Et₃N treatments.

7. In the support information (e.g., page s6, line 77, page s9, line 120), there are errors in the remarks of the nuclear magnetic instrument information, please verify and correct.
 - We sincerely appreciate the reviewer's thorough reading. After carefully reviewing the supplementary information, we have corrected all errors regarding the remarks on nuclear magnetic instrument information.

8. Overall, considering the demonstrated results with special phenomena and insightful mechanisms, this work is recommended for publication in Nature Communications.
 - We are deeply grateful for the reviewer's recommendation for our work to be published in Nature Communications.

In the end, we would like to express our sincere gratitude to the reviewers once again for the invaluable assistance. The reviewers thoroughly reviewed our manuscript and proposed comments comprehensively including experimental design, mechanistic understanding, future prospects, and manuscript writing. These constructive comments have not only stimulated deep reflection but also provided fresh insights into our research on mechanically interlocked networks. Inspired by these perspectives, we are motivated to devise new experiments in our future endeavors, which will compellingly support our conclusions and further solidify our research.

REVIEWERS' COMMENTS

Reviewer #1 (Remarks to the Author):

In the revised version, the concerns have been well addressed. It can be accepted by Nature Communications.

Reviewer #3 (Remarks to the Author):

The authors have revised the manuscript according to the comments of the reviewers. I recommend its publication in Nat Commun in this current version.

Response to Reviewers' Comments

Reviewer #1 Comment In the revised version, the concerns have been well addressed. It can be accepted by Nature Communications.

Response: We thank the reviewer for their positive assessment of the revised manuscript and for supporting its publication.

Reviewer #3 Comment The authors have revised the manuscript according to the comments of the reviewers. I recommend its publication in Nat. Commun. in this current version.

Response: We thank the reviewer for their valuable time and for supporting publication of the revised manuscript.